# Learning Joint Morpho-Molecular Tissue Representations with a Multimodal Transformer

**Adriano Martinelli**
Biomedical Data Science Center, CHUV
adriano.martinelli@chuv.ch

**Bernd Illing**
Biomedical Data Science Center, CHUV
bernd.illing@chuv.ch

**Isinsu Katircioglu**
Biomedical Data Science Center, CHUV
isinsu.katircioglu@chuv.ch

**Alice Driessen**
kaiko.ai
alice.driessen@kaiko.ai

**Fei Tang**
kaiko.ai
fei.tang@kaiko.ai

**Raphael Gottardo**
Biomedical Data Science Center, CHUV
raphael.gottardo@chuv.ch

**Marianna Rapsomaniki**
Biomedical Data Science Center, CHUV
marianna.rapsomaniki@chuv.ch

## Abstract

Understanding how molecular programs are embedded within tissue morphology is a central challenge in spatial biology. While vision transformer (ViT) foundation models capture rich histological structure and spatial transcriptomics (ST) provides molecular context, existing multimodal approaches largely rely on contrastive alignment and do not directly learn joint morpho-molecular representations. We introduce an early-fusion multimodal transformer that integrates subcellular Xenium transcript readouts directly into the ViT token stream, enabling fine-grained cross-modal interaction without cell segmentation. We evaluate our approach on a gene prediction task, predicting held-out genes from a targeted Xenium panel given histology and a core gene set. Across a comprehensive benchmark of unimodal baselines and *vanilla* late-fusion variants, early fusion achieves substantial improvements in gene expression prediction. We further show that performance gains are driven primarily by spatially aligned, token-level transcript representations rather than fusion timing alone. With appropriate transcript tokenization, late fusion can perform on par with early fusion, which explains the limitations observed in prior CLIP-style models. Our results highlight expressive, spatially grounded fusion as a key ingredient for multimodal representation learning in spatial biology.

## 1 Introduction

Tissue organization arises from the coordinated arrangement of diverse cell types, each with distinct morphologies, phenotypes, and molecular programs (Kashyap et al., 2022). Understanding how molecular states occur within tissues is central to many biological and clinical questions, from tumor–immune interactions to spatial signaling programs. Histopathology, based on hematoxylin and eosin (H&E) staining of whole-slide images (WSIs), provides a rich view of tissue morphology across tissue scales. Large-scale self-supervised pretraining of vision transformers (ViTs) on millions of H&E WSIs has led to general-purpose foundation models (FMs) (Chen et al., 2024; Vorontsov et al., 2024) that excel across diverse pathology-related tasks. Morphology alone, however, does not directly capture the molecular programs that determine cell identity and functional state. Emerging spatial transcriptomics (ST) technologies promise to fill this gap: imaging-based ST such as Xenium (10x Genomics) map gene expression transcripts within tissues at subcellular resolution (Bressan et al., 2023), potentially exposing molecular mechanisms in health and disease (Janesick et al., 2023).

Recent work has begun to explore unified representations combining morphology and gene expression. One line of research focuses on training deep neural networks to *predict* gene expression from H&E representations (e.g., HisToGene (Pang et al., 2021), ST-Net (He et al., 2020) and DeepSpot (Nonchev et al., 2025)). More recent approaches (e.g., OmiCLIP (Chen et al., 2025), PathOmCLIP (Lee et al., 2024) and BLEEP (Xie et al., 2023)) adopt a CLIP-style paradigm (Radford et al., 2021), training dual encoders to *align* H&E images and ST data in a shared latent space. Finally, graph-based FMs for ST have also emerged (Novae Blampey et al. (2025)), integrating H&E and ST data for various tasks such as spatial domain analysis. However, existing CLIP-based models align morphology and transcriptomics as separate modalities in a shared latent space, rather than jointly modeling them during representation learning: ST measurements act as supervisory signals that shape the image encoder through contrastive alignment, but the learned encoders remain unimodal at inference and do not provide a joint morpho-molecular representation. Cross-modal interactions are restricted to aggregated feature spaces, limiting the ability to capture fine-grained dependencies between local tissue morphology and gene expression. In a recent large-scale benchmark (Gindra et al., 2025), several CLIP-based approaches were compared on a curated version of the HEST-1k dataset (Jaume et al., 2024) for gene expression prediction, reporting that, surprisingly, image encoders outperform contrastive pretraining with gene expression encoders. This suggests that contrastive alignment alone may fail to leverage molecular information beyond what is already encoded in pathology FMs. This is further complicated by the fact that imaging-based ST rely on disparate gene panels that greatly differ across tissues and experiments, introducing batch effects and constraining dataset integration and multimodal learning.

In this work, we introduce the first early-fusion multimodal transformer architecture for joint representation learning of histology and spatial transcriptomics (Xenium). Our approach directly exploits subcellular transcript readouts via a token-level early-fusion mechanism that injects transcript-derived tokens *directly* into the ViT token sequence prior to any shared processing, circumventing the need for cell segmentation, a time-consuming and error-prone process, further complicated by transcript diffusion (Bilous et al., 2025). This architecture enables all transformer layers to operate on *jointly contextualized morphology–transcript representations*, rather than restricting cross-modal interaction to aggregated feature spaces. We evaluate different flavors of our fusion model on the task of predicting an add-on panel of held-out genes from the core Xenium panel, thus forcing the model to learn biologically coherent representations that reflect gene–gene relationships and morpho-molecular coupling. We systematically evaluate model performance against several unimodal FMs and late fusion baselines and examine how the granularity of cross-modal interaction influences performance. Our results show that joint token-level processing consistently improves gene expression prediction, highlighting that, in contrast to conclusions from recent benchmarks Gindra et al. (2025), fusing histopathology and spatial transcriptomics in a biologically grounded way captures joint morpho-molecular representations, enabling accurate prediction of unmeasured genes.

## 2 METHODS

We tile each WSI into $512 \times 512$ pixel tiles with a stride of 256 at a resolution of 0.5 μm per pixel, yielding a total of 338,662 tiles. The Xenium panel is partitioned into a *core* panel of $G_{\text{core}}$ genes and an *add-on* panel of $G_{\text{add}}$ genes. For each spatial location (tile), given available inputs (H&E and/or transcripts from the core set), we predict the target add-on gene expression vector $\mathbf{y} \in \mathbb{R}^{G_{\text{add}}}$, as shown in Figure 1. We define targets by aggregating add-on transcripts within each tile and applying a log1p transform to obtain $\mathbf{y}$. On the image side, each tile is resized to $224 \times 224$ pixels and encoded with the vision token encoder of a ViT, which tokenizes the tile into $N$ patch tokens (plus an optional classification token) and outputs token embeddings $\mathbf{H}_{\text{img}} \in \mathbb{R}^{(N+1) \times d}$, with patch-token embeddings denoted by $\mathbf{h}_{\text{img}}^{(i)} \in \mathbb{R}^d$ for $i \in \{1, \ldots, N\}$. On the transcript side, we consider two core representations: (i) *tile-level aggregation*, yielding a single vector $x_{\text{core}}$ per tile (analogous to the target definition), and (ii) *token-level aggregation*, yielding a sequence aligned to the ViT patch grid. Concretely, using patch coordinates in the original image space, we pool core-panel transcripts into each patch $i$ to obtain $\mathbf{x}_{\text{core}}^{(i)} \in \mathbb{R}^{G_{\text{core}}}$, which is spatially aligned with $\mathbf{h}_{\text{img}}^{(i)}$; we embed these features via $\mathbf{h}_{\text{core}}^{(i)} = \text{MLP}(\mathbf{x}_{\text{core}}^{(i)})$ and collect them in $\mathbf{H}_{\text{core}} \in \mathbb{R}^{N \times d}$.

We benchmark unimodal and multimodal baselines. For morphology (**Morph-only**) settings, we use pretrained H&E FMs as frozen feature extractors, including CONCH Lu et al. (2024), Phikon

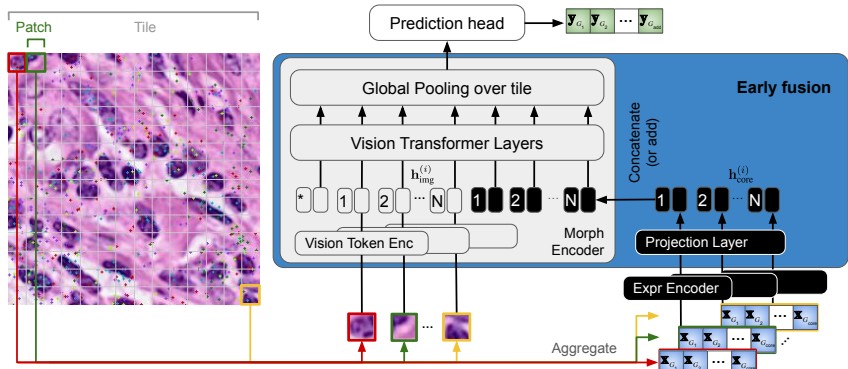

Figure 1: **Model architecture.** H&E tiles and core panel Xenium transcripts are spatially aligned and tokenized on the ViT grid. H&E tokens are encoded by the *Vision Token Encoder* of the ViT (*Morph Encoder*). Patch-aggregated core panel transcripts (blue) are encoded (*Expr Encoder*) and projected to the dimensionality of the encoded vision tokens. Encoded expression tokens are then injected into the ViT before the remaining ViT layers to enable multimodal encoding. A linear prediction head then predicts tile-aggregated gene expression of the add-on panel (green).

Filiot et al. (2024), and the OmiCLIP/Loki image encoder Chen et al. (2025), and additionally evaluate a ViT-S backbone from `timm` under frozen feature extraction and end-to-end fine-tuning. For expression-only (**Expr-only**) settings, we train a shallow 1-layer MLP with 32 units that maps tile-level ($\mathbf{x}_{\text{core}}$) or token-level ($\mathbf{x}_{\text{core}}^i$) core expression to add-on predictions $\mathbf{y}$, and, in late fusion experiments, we also consider a frozen Geneformer model (gf-12L-38M-i4096) Theodoris et al. (2023) as an alternative transcript encoder. For **Late fusion**, modalities are processed independently and combined at the embedding level: letting $\mathbf{z}_{\text{img}} \in \mathbb{R}^d$ denote a pooled image embedding (e.g., CLS token) and $\mathbf{z}_{\text{core}} \in \mathbb{R}^d$ a transcript embedding of core expression, we test concatenation and elementwise addition. As a **tokenized late fusion** variant, we compute token-level transcript embeddings $\mathbf{h}_{\text{core}}^{(i)} = \text{MLP}(\mathbf{x}_{\text{core}}^{(i)})$, average pool them into $\mathbf{z}_{\text{core}} = \frac{1}{N} \sum_i \mathbf{h}_{\text{core}}^{(i)}$, and then fuse as above, isolating the effect of tokenized (instead of tile-pooled) transcript features (see Figure A.6). For **Early fusion**, we integrate transcript tokens into the ViT token stream before the transformer stack: given $\mathbf{H}_{\text{img}}$ and $\mathbf{H}_{\text{core}}$, we test elementwise addition of aligned image and transcript tokens and concatenation of the two token sequences, and pass the resulting fused sequence (plus an optional CLS token) through the ViT transformer layers. In all cases, a linear head maps the final representation from either late or early fusion to add-on predictions $\mathbf{y}$. Parameter counts for all models are given in Table A.3.

All models are trained to predict log1p-transformed add-on gene expression using MSE loss over add-on genes and are evaluated per gene and averaged across genes using correlation-based metrics (Pearson, Spearman) and MSE, consistent with prior evaluations (Chen et al., 2025; Gindra et al., 2025). As a simple transcript-only reference, we include a **Min-MAE baseline** that predicts each target gene $t \in G_{\text{add}}$ by copying the expression of the single core gene $s^*(t) \in G_{\text{core}}$ that minimizes training-set MAE, $s^*(t) = \arg\min_{s \in \{1,\ldots,G_{\text{core}}\}} \text{MAE}(y_{\cdot,t}, x_{\cdot,s})$, which is equivalent to a constrained linear model with exactly one nonzero coefficient per target (fixed to 1). Finally, we perform **CLIP pretraining**, to further compare tile-level vs. token-level core transcript representations: in the tile-level variant, we contrast the image embedding $\mathbf{h}_{\text{img}}$ against a single tile-aggregated core-transcript embedding $\mathbf{h}_{\text{core}}$, whereas in the token-level variant we contrast $\mathbf{h}_{\text{img}}$ against the mean-pooled token-level embeddings, testing whether preserving within-tile transcript heterogeneity during pretraining improves representation quality. All experiments were performed on an internal cancer dataset of 159 patients with paired H&E and Xenium measurements using the core Human Immuno-Oncology Panel ($G_{\text{core}} = 380$ genes) and an add-on custom panel ($G_{\text{add}}=100$ genes). The dataset will be published in a separate publication.

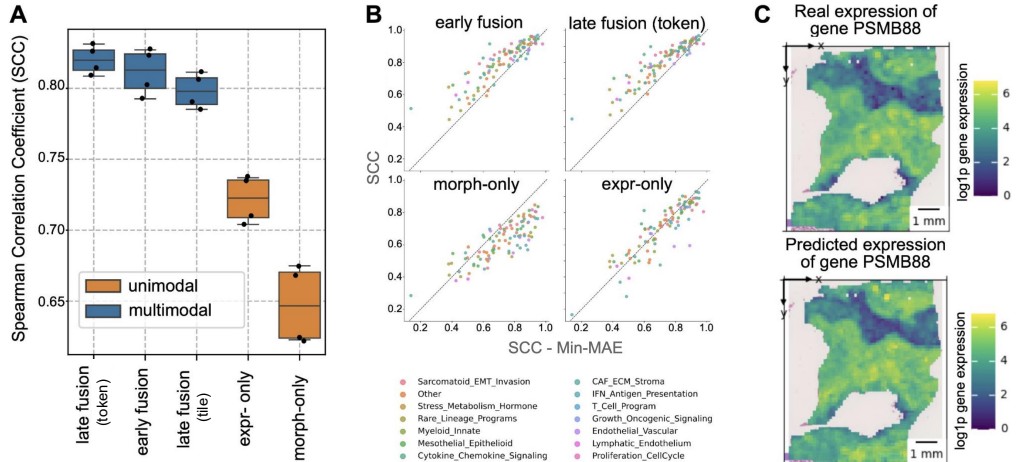

Figure 2: **Multimodal gene expression prediction.** (**A**) Predictive performance (mean Spearman correlation coefficient (SCC) across add-on genes) across 4-fold, group-stratified cross-validation comparing unimodal baselines (*morph-only, expr-only*) and multimodal fusion strategies (*early fusion, late fusion* with tile-level pooling *(tile)* and token-level pooling *(token)*). (**B**) SCC comparison to Min-MAE baseline (see methods) for one representative split across all target genes, annotated by gene programs. (**C**) Example tile-level reconstruction for a representative gene using early fusion, illustrating calibration and dynamic range of predictions.

## 3 RESULTS

A comprehensive benchmark of all model variants is reported in Table A.1, spanning image encoders (pretrained H&E FMs and ViT-S), expression encoders, transcript aggregation/tokenization strategies, fusion strategies and stages, and training regimes. We observe that, across all configurations, our **multimodal fusion models outperform all unimodal baselines** (Figure 2A), with a 26.5% and a 13% improvement in Spearman correlation coefficient over the best-in-class Morph-only and Expr-only baselines, respectively. We also observe that **token-level transcript representation is a key driver**: token-level early fusion slightly outperforms *vanilla* late fusion that uses tile-level transcripts (Figure 2A; Table A.1). However, the main determinant is not fusion timing *per se*, but the granularity of the transcript representation: when late fusion is augmented with spatially aligned, token-level transcript pooling, the gap closes and *tokenized* late fusion performs on par with early fusion. This points to *spatially aligned transcript summarization* as a major contributor to performance. To assess this, we also performed CLIP-style pretraining (Table A.2) and confirm that token-level transcript aggregation improves retrieval compared to tile-level aggregation (recall@1 $0.062 \pm 0.004$ vs. $0.041 \pm 0.003$). Finally, across all experiments, **lightweight, task-specific encoders consistently outperform frozen H&E FM embeddings**: in particular, a low-capacity ViT-S trained on our data (with a linear head) and an MLP for expression encoding achieve higher correlations than linear probes on frozen CONCH, Phikon, or OmiCLIP/Loki features (Table A.1). We also observe that **both early and late fusion models clearly outperform the Min-MAE baseline** across all genes of all modules (Figure 2B). Conversely, the Expr-only model performs comparably to the Min-MAE baseline, suggesting that its performance arises from simple cross-panel correlations (Figure A.1, Figure A.2), and the Morph-only model is clearly outperformed by the Min-MAE baseline across almost all genes. When plotted spatially, our gene expression prediction results closely mirror the real measurements (Figures 2C and A.3), and reveal rich spatial morpho-molecular structures of the underlying tumor microenvironment (Figure A.4).

## 4 DISCUSSION

In this work, we show that jointly modeling histopathology and spatial transcriptomics through multimodal transformers enables more effective learning of morpho-molecular representations than existing contrastive approaches. By fusing subcellular Xenium transcript readouts directly into the ViT token stream at the patch level, our model captures fine-grained spatial dependencies between

tissue morphology and gene expression without requiring cell segmentation. Across a comprehensive benchmark, we demonstrate that multimodal fusion consistently outperforms unimodal image- or expression-only models for predicting unmeasured genes, highlighting the complementary nature of morphological and molecular information. Our results further reveal that the granularity of transcript representation—specifically, spatially aligned, token-level summarization—is a primary driver of performance. Together, these findings suggest that limitations observed in prior CLIP-style models arise not necessarily from the absence of molecular signal, but from insufficiently expressive integration mechanisms, which can be mitigated by tokenizing transcripts. By framing gene prediction as a principled pretraining task on targeted panels, our approach provides a biologically grounded path toward scalable multimodal foundation models for spatial transcriptomics.

**Limitations and future work.** We are currently working on a number of baselines and extensions, including: (i) using our approach on a target panel aligned with DeepSpot (Nonchev et al., 2025). Early results indicate consistent performance, suggesting robustness to panel choice. (ii) running preliminary transfer experiments with the frozen CLIP-pretrained ViT-S backbone. Linear probing suggests that token-level CLIP embeddings outperform tile-level pooled embeddings (CLIP-token > CLIP-tile) for gene prediction. Our evaluation is currently limited to a single internal dataset and lightweight fusion mechanisms, and predictions are assessed only at a tile level. We are currently working on extending to multi-center cohorts such as the HEST-1k dataset, exploring more expressive early fusion (e.g., gated or cross-attention mechanisms), higher-resolution gene prediction, and whole-slide prediction tasks. Finally, systematically curating target panels to reduce core–add-on redundancy, and introducing more challenging tasks such as cell type abundance prediction, may provide a more stringent stress test for multimodal fusion strategies.

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

# A  APPENDIX

## A.1  DATA

**H&E and Xenium alignment.** We register the H&E whole-slide image (WSI) to the Xenium coordinate system using the DAPI channel and manually selected landmark correspondences. These landmarks are used to estimate an affine transformation that maps Xenium transcript coordinates into H&E pixel space. To account for local tissue deformations, we partition each tissue section into regions and estimate a separate affine transform per region. Finally, we crop the WSI to the transcript-containing area to restrict downstream tiling to regions with Xenium measurements. We tile each WSI into 512×512 pixel tiles with stride of 256 pixels at a resolution of $0.5\mu m$ per pixel, yielding a total of 344,328 tiles. We filter tiles to contain at least 1000 transcripts to ensure enough transcriptomic signal for models that rely on it (Figure A.5). This results in our final dataset with 338,662 tiles.

## A.2  IMPLEMENTATION DETAILS.

**MSE loss.**

$$\mathcal{L}(\hat{\mathbf{y}}, \mathbf{y}) = \frac{1}{G_{\text{add}}} \sum_{g=1}^{G_{\text{add}}} (\hat{y}_g - y_g)^2.$$

**Training protocol.** All models were implemented in PyTorch and Lightning to ensure reproducibility and consistency across experiments. Training was performed on NVIDIA A100 GPUs (40 GB memory) using mixed-precision arithmetic. Each model configuration was trained for up to 35 epochs (approximately 2–7 hours wall-clock time, depending on model size), with early stopping based on validation performance.

Optimization used AdamW. Unless stated otherwise, we used a learning rate of $10^{-4}$ for the prediction head and $10^{-5}$ for backbone parameters, weight decay of $10^{-3}$, and a global batch size between 256 and 512 depending on memory requirements. Gradients were clipped to a maximum norm of 1.0. We used a cosine learning-rate schedule with a linear warmup of 5 epochs, followed by cosine annealing to a minimum learning rate of $10^{-6}$.

All experiments used deterministic seeding. Model selection was based on validation mean squared error, with early stopping patience of 10 epochs. Data were split into four fixed training/validation/test partitions, as shown in Table A.4. Target gene expression values were log1p-transformed prior to training. Hyperparameters, evaluation metrics (MAE, MSE/RMSE, Pearson and Spearman correlation), learning-rate schedules, and checkpoint metadata were logged to an experiment tracking system.

Table A.1: Hyperparameter settings for all evaluated models. Settings 0–1 correspond to *early fusion*, settings 3–26 to *late fusion*, settings 27–31 to *Morph-only*, and settings 32–33 to *Expr-only*. Entries marked with {*} indicate runs that did not converge.

| setting | fusion_stage | fusion_strategy | morph_encoder | freeze_morph | expr_encoder | freeze_expr | expr_aggregation | spearman ↑ | pearson ↑ | mse ↓ |
|---|---|---|---|---|---|---|---|---|---|---|
| 0 | early | add | vit_s | False | mlp | False | token | **0.811±0.017** | **0.820±0.013** | **0.705±0.030** |
| 1 | early | add | vit_s | True | mlp | False | token | 0.771±0.014 | 0.777±0.011 | 0.862±0.018 |
| 2 | early | concat | vit_s | False | mlp | False | token | 0.809±0.017 | 0.818±0.013 | 0.768±0.058 |
| 3 | late | add | conch | True | gf | True | tile | 0.727±0.017 | 0.747±0.012 | 0.979±0.014 |
| 4 | late | add | conch | True | mlp | False | tile | 0.752±0.012 | 0.763±0.008 | 0.942±0.028 |
| 5 | late | add | loki | True | gf | True | tile | 0.696±0.021 | 0.718±0.013 | 1.107±0.027 |
| 6 | late | add | loki | True | mlp | False | tile | 0.769±0.036 | 0.776±0.033 | 0.873±0.090 |
| 7 | late | add | phikon | True | gf | True | tile | 0.738±0.019 | 0.754±0.014 | 0.966±0.012 |
| 8 | late | add | phikon | True | mlp | False | tile | 0.787±0.013 | 0.796±0.011 | 0.799±0.011 |
| 9 | late | add | vit_s | False | gf | True | tile | 0.684±0.023 | 0.700±0.017 | 1.177±0.018 |
| 10 | late | add | vit_s | False | mlp | False | token | 0.728±0.018 | 0.741±0.015 | 1.057±0.040 |
| 11 | late | add | vit_s | False | mlp | False | tile | 0.773±0.016 | 0.778±0.014 | 0.887±0.029 |
| 12 | late | add | vit_s | True | gf | True | tile | 0.715±0.019 | 0.735±0.012 | 1.030±0.022 |
| 13 | late | add | vit_s | True | mlp | False | tile | 0.797±0.012 | 0.805±0.009 | 0.767±0.007 |
| 14 | late | add | vit_s | True | mlp | False | token | 0.813±0.010 | 0.819±0.007 | 0.706±0.008 |
| 15 | late | concat | conch | True | gf | True | tile | 0.738±0.018 | 0.756±0.013 | 0.950±0.011 |
| 16 | late | concat | conch | True | mlp | False | tile | 0.778±0.026 | 0.788±0.023 | 0.834±0.076 |
| 17 | late | concat | loki | True | gf | True | tile | 0.707±0.022 | 0.728±0.015 | 1.075±0.030 |
| 18 | late | concat | loki | True | mlp | False | tile | 0.781±0.020 | 0.788±0.018 | 0.825±0.034 |
| 19 | late | concat | phikon | True | gf | True | tile | 0.742±0.019 | 0.758±0.014 | 0.953±0.012 |
| 20 | late | concat | phikon | True | mlp | False | tile | 0.784±0.013 | 0.793±0.010 | 0.813±0.009 |
| 21 | late | concat | vit_s | False | gf | True | tile | 0.694±0.032 | 0.712±0.026 | 1.141±0.036 |
| 22 | late | concat | vit_s | False | mlp | False | tile | 0.740±0.019 | 0.752±0.016 | 1.011±0.031 |
| 23 | late | concat | vit_s | False | mlp | False | token | 0.786±0.016 | 0.791±0.013 | 0.829±0.021 |
| 24 | late | concat | vit_s | True | gf | True | tile | 0.722±0.018 | 0.742±0.012 | 1.007±0.019 |
| 25 | late | concat | vit_s | True | mlp | False | tile | 0.798±0.013 | 0.806±0.010 | 0.761±0.006 |
| 26 | late | concat | vit_s | True | mlp | False | token | **0.820±0.010** | **0.826±0.008** | **0.673±0.010** |
| 27 | - | - | conch | True | - | False | - | 0.626±0.017 | 0.631±0.018 | 1.359±0.026 |
| 28 | - | - | loki* | True | - | False | - | 0.354±0.036 | 0.374±0.032 | 6.679±0.279 |
| 29 | - | - | phikon | True | - | False | - | 0.633±0.024 | 0.637±0.022 | 1.367±0.031 |
| 30 | - | - | vit_s | False | - | False | - | **0.648±0.028** | **0.654±0.026** | **1.332±0.037** |
| 31 | - | - | vit_s | True | - | False | - | 0.587±0.020 | 0.592±0.017 | 1.495±0.036 |
| 32 | - | - | - | False | gf | True | tile | 0.622±0.044 | 0.662±0.029 | 1.314±0.128 |
| 33 | - | - | - | False | mlp | False | token | **0.722±0.017** | **0.722±0.013** | **1.055±0.026** |
| min-mae | - | - | - | - | - | - | - | 0.724±0.010 | 0.731±0.006 | 1.144±0.044 |

gf: Geneformer gf-12L-38M-i4096
vit_s: vit_small_patch16_224.augreg_in21k

Table A.2: **CLIP pretraining retrieval performance.** Image–transcript retrieval results (recall@1 and recall@5; mean±std across runs) for different core-transcript aggregation strategies.

| | expr_encoder | morph_encoder | expr_aggregation | recall@1 | recall@5 |
|---|---|---|---|---|---|
| 0 | mlp | vit_s | token | 0.062±0.004 | 0.202±0.009 |
| 1 | mlp | vit_s | tile | 0.041±0.003 | 0.151±0.011 |

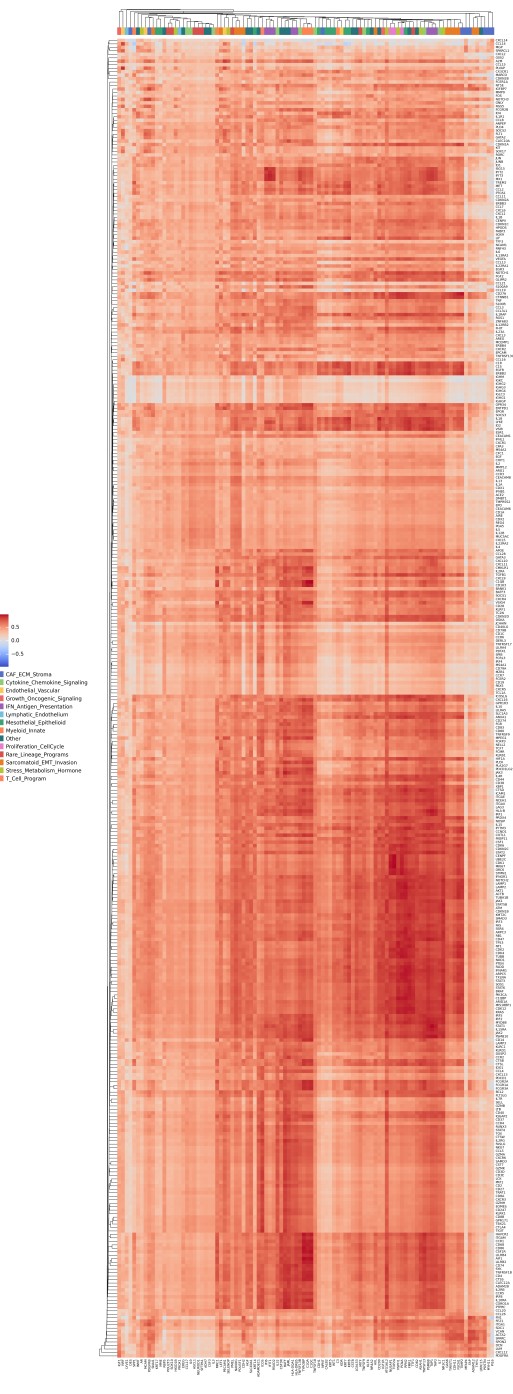

Figure A.1: **Source–target gene correlation matrix.** Correlation between source genes (rows) and target genes (columns), highlighting targets that are strongly correlated with the source panel (high redundancy) versus targets that are less recoverable from correlated proxies.

Table A.3: Model complexities (parameter counts) for the evaluated architectures.

| | head total | head trainable | backbone total | backbone trainable | fusion_strategy | fusion_stage | morph_encoder_na | freeze_morph | expr_encoder | freeze_expr |
|---|---|---|---|---|---|---|---|---|---|---|
| fusion-early | 38500 | 38500 | 21694560 | 28896 | concat | early | vit_small_patch16_224 | True | mlp | False |
| fusion-early | 38500 | 38500 | 21694560 | 28896 | add | early | vit_small_patch16_224 | True | mlp | False |
| fusion-early | 38500 | 38500 | 21694560 | 21694560 | concat | early | vit_small_patch16_224 | False | mlp | False |
| fusion-early | 38500 | 38500 | 21694560 | 21694560 | add | early | vit_small_patch16_224 | False | mlp | False |
| fusion-late | 153700 | 153700 | 638493665 | 43488 | concat | late | loki | True | mlp | False |
| fusion-late | 76900 | 76900 | 638493665 | 43488 | add | late | loki | True | mlp | False |
| fusion-late | 204900 | 204900 | 303405024 | 53216 | concat | late | phikon | True | mlp | False |
| fusion-late | 102500 | 102500 | 303405024 | 53216 | add | late | phikon | True | mlp | False |
| fusion-late | 153700 | 153700 | 306152672 | 43488 | concat | late | conch_v1.5 | True | mlp | False |
| fusion-late | 76900 | 76900 | 306152672 | 43488 | add | late | conch_v1.5 | True | mlp | False |
| fusion-late | 76900 | 76900 | 21864576 | 198912 | concat | late | vit_small_patch16_224 | True | geneformer | True |
| fusion-late | 38500 | 38500 | 21864576 | 198912 | add | late | vit_small_patch16_224 | True | geneformer | True |
| fusion-late | 153700 | 153700 | 638848001 | 397824 | concat | late | loki | True | geneformer | True |
| fusion-late | 76900 | 76900 | 638848001 | 397824 | add | late | loki | True | geneformer | True |
| fusion-late | 204900 | 204900 | 303882240 | 530432 | concat | late | phikon | True | geneformer | True |
| fusion-late | 102500 | 102500 | 303882240 | 530432 | add | late | phikon | True | geneformer | True |
| fusion-late | 153700 | 153700 | 306507008 | 397824 | concat | late | conch_v1.5 | True | geneformer | True |
| fusion-late | 76900 | 76900 | 306507008 | 397824 | add | late | conch_v1.5 | True | geneformer | True |
| morph | 38500 | 38500 | 21667584 | 21667584 | - | - | vit_small_patch16_224 | False | - | False |
| morph | 38500 | 38500 | 21667584 | 1920 | - | - | vit_small_patch16_224 | True | - | False |
| morph | 76900 | 76900 | 638454017 | 3840 | - | - | loki | True | - | False |
| morph | 102500 | 102500 | 303356928 | 5120 | - | - | phikon | True | - | False |
| morph | 76900 | 76900 | 306113024 | 3840 | - | - | conch_v1.5 | True | - | False |
| expr | 3300 | 3300 | 14464 | 14464 | - | - | - | False | mlp | False |
| expr | 51300 | 51300 | 2560 | 2560 | - | - | - | False | geneformer | True |

Table A.4: **Train/validation/test split statistics.** Number of tiles assigned to the training (*fit*), validation (*val*), and test (*test*) partitions for each outer split (inner=0, seed=0). The dataset contains 159 samples in total and 338,662 tiles overall; tile counts per partition vary slightly across splits due to sample-level partitioning and differences in the number of tiles per sample.

| split | fit | test | val | num_samples | num_tiles |
|---|---|---|---|---|---|
| outer=0-inner=0-seed=0 | 192580 | 86493 | 59589 | 159 | 338662 |
| outer=1-inner=0-seed=0 | 188090 | 84451 | 66121 | 159 | 338662 |
| outer=2-inner=0-seed=0 | 191184 | 83058 | 64420 | 159 | 338662 |
| outer=3-inner=0-seed=0 | 191368 | 84660 | 62634 | 159 | 338662 |

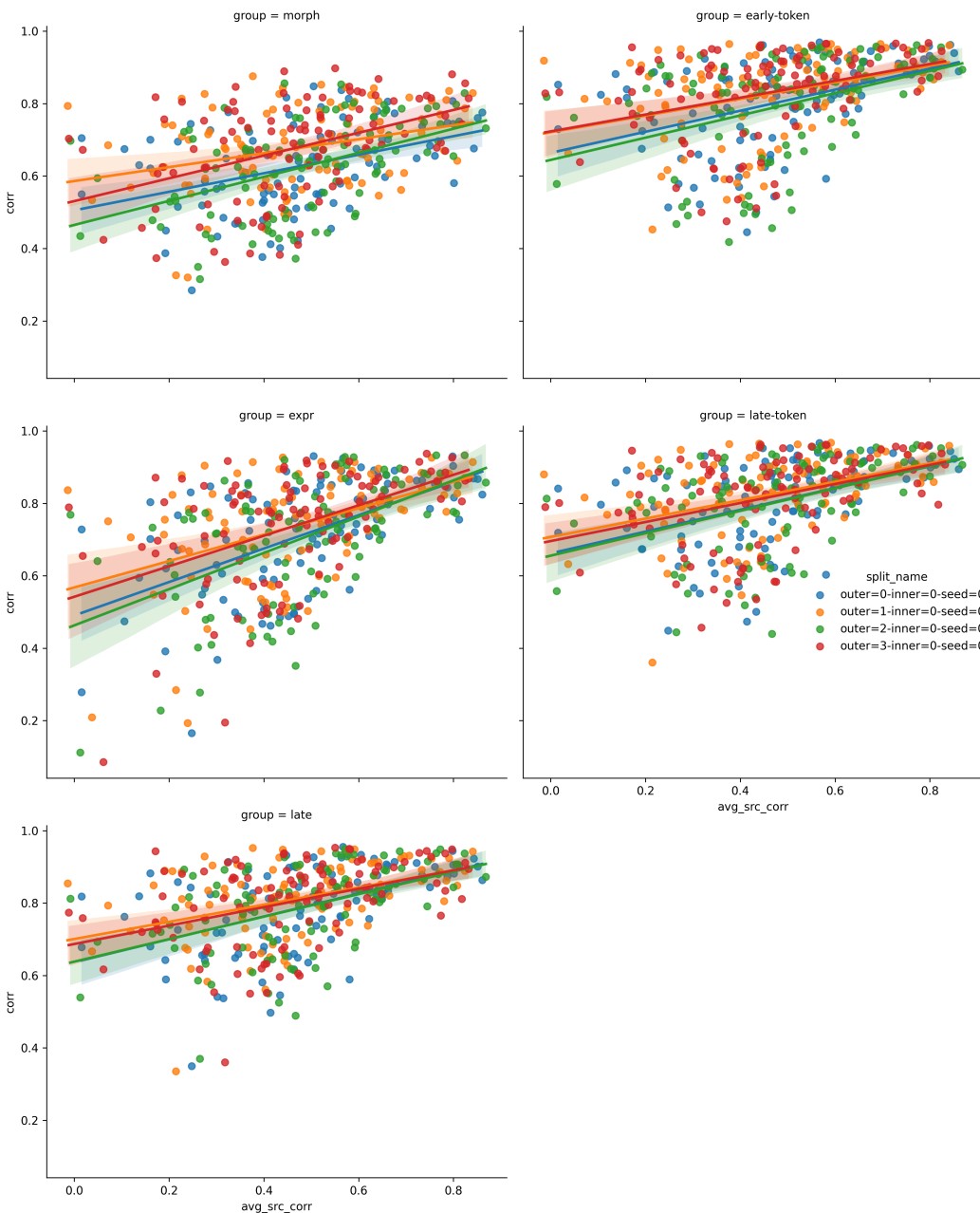

Figure A.2: **Predictability vs. redundancy with the source panel.** Per-target prediction performance as a function of `avg_src_corr`, the average correlation of a target gene to all genes in the source (core) panel. Higher `avg_src_corr` indicates stronger redundancy with the source panel and is associated with improved predictability.

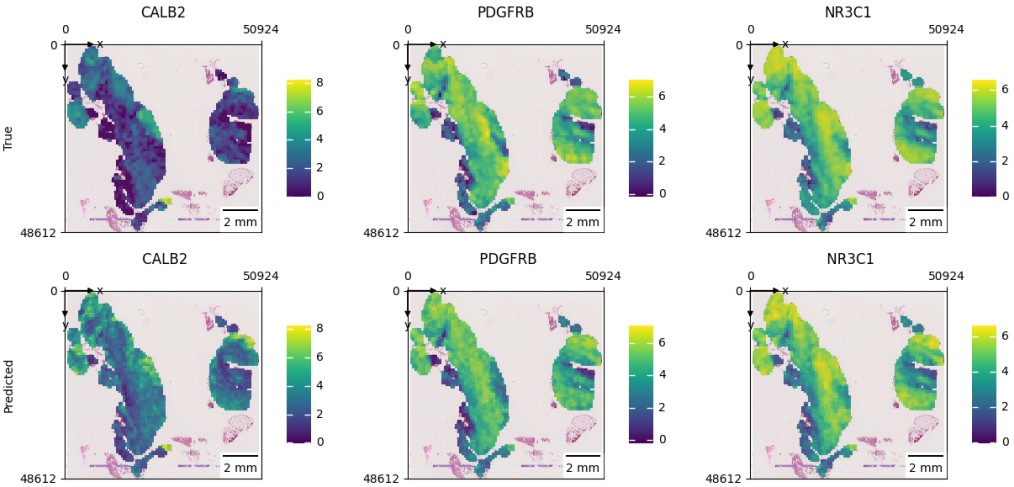

Figure A.3: **Tile-level predictions on one example sample.** Predicted versus true expression values per tile for the late fusion model. Three representative genes are shown to span the within-sample performance distribution: CALB2 (lowest), PDGFRB (median), and NR3C1 (highest).

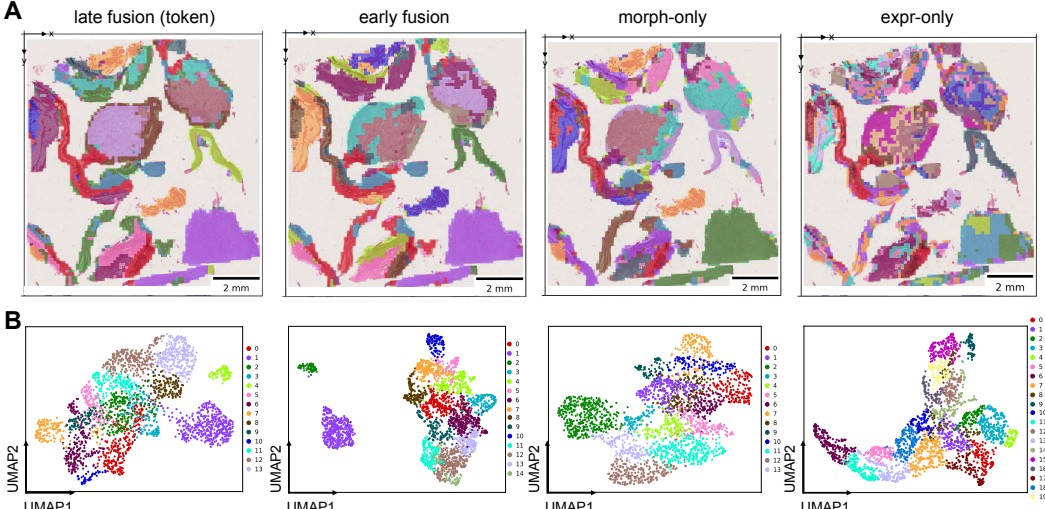

Figure A.4: **Learned embeddings capture rich morpho-molecular descriptions of the tumor microenvironment.** (**A**) Leiden clustering of image/tile embeddings for a single sample across models. For fairness, embeddings are reduced by PCA to 31 dimensions (matching the MLP bottleneck capacity; MLP uses 32) and clustered with fixed Leiden resolutions (0.5 and 1.0) without per-model tuning. Expr-only yields granular, noisy clusters; morph-only produces larger coherent regions (potential under-clustering); early fusion better reflects transcript-driven heterogeneity while remaining spatially coherent; late fusion closely mirrors morph-only. Cluster colors are not aligned across methods. (**B**) Corresponding UMAP visualizations of the embeddings colored by the Leiden clusters from (A), summarizing global neighborhood structure under identical dimensionality reduction and clustering settings.

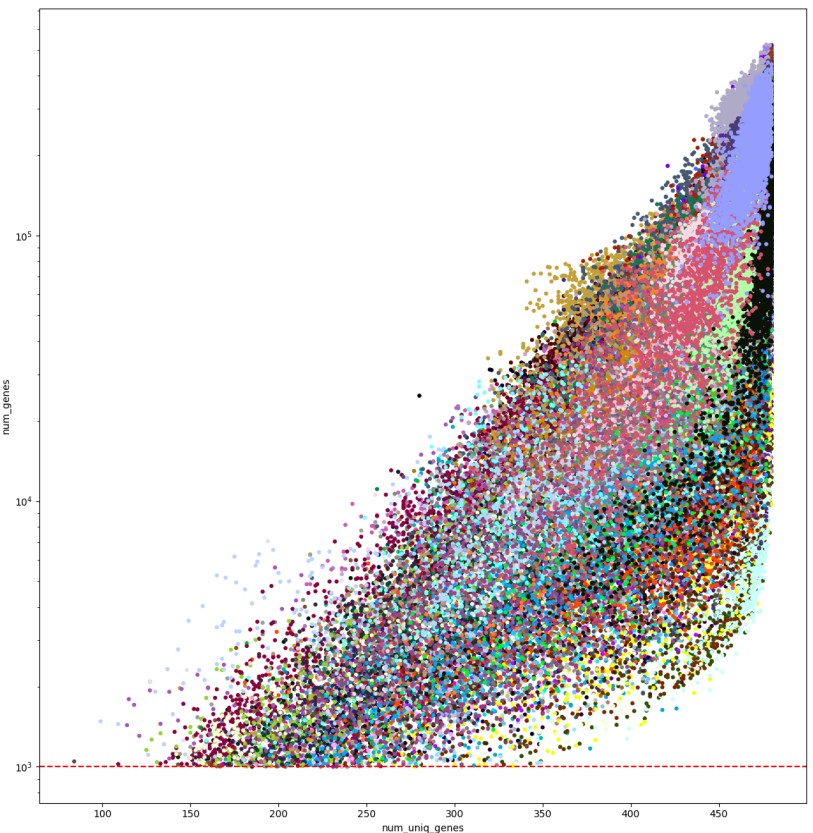

Figure A.5: **Unique transcripts vs. detected genes per tile.** Relationship between the number of unique transcript molecules and the number of genes detected per tile (log scale).

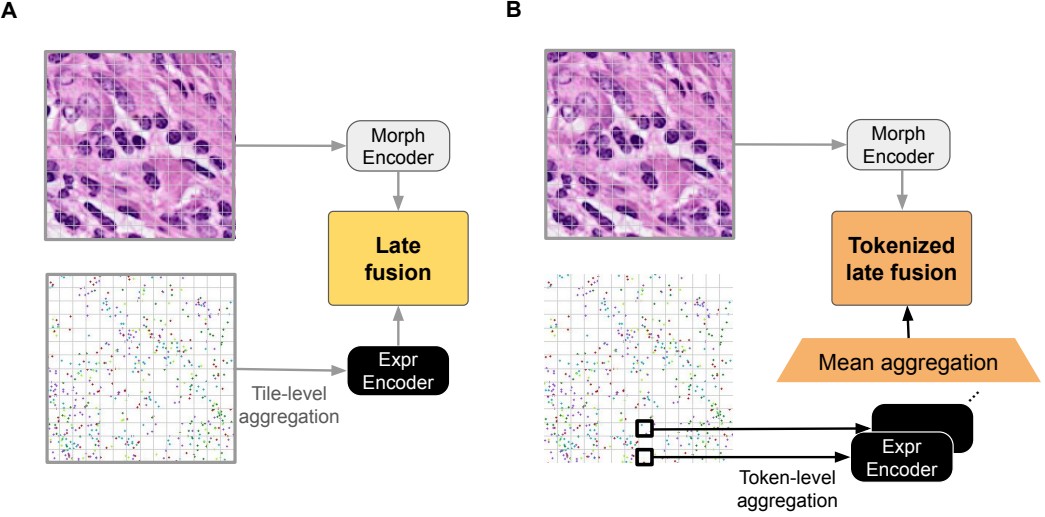

Figure A.6: **Late fusion variants.** (**A**) *Vanilla* late fusion aggregates transcripts at the tile level. Morphology and expression encoders then produce tile-level embeddings, which are subsequently combined. (**B**) Tokenized late fusion, in contrast, aggregates transcripts at the patch level to produce patch-level expression embeddings. These are mean-aggregated and combined with the tile-level morphology embeddings.

