# OpenReview forum: "Learning Joint Morpho-Molecular Tissue Representations with a Multimodal Transformer"
_ICLR.cc/2026/Workshop/LMRL — ICLR 2026 Workshop LMRL Poster_

### Official Review · Reviewer_22Dz · 2026-02-24
**Comprehensive benchmarks on fusion strategies**

**Rating:** 9
**Confidence:** 3

**Review:**

Authors suggest the early fusion approach for H&E imaging and Xenium expression modalities, namely, tokenization of the patch expressions that are fed into a transformer block along with image tokens. Authors have conducted benchmarks across comprehensive baselines and hyperparameters. They found that not only does their early fusion approach outperform all unimodal baselines, but also the late fusion approaches, where token-level expression embeddings are spatially matched with the token-level image embeddings, performed on par or better than their early-fusion approach. Thereby, this work contributes both in suggesting a novel architecture and providing comprehensive benchmarks, helping elucidate key components of spatial multimodal alignment.

Many of the limitations or future directions are already well discussed, including application to more datasets and going beyond tile-level prediction. I would like to see more discussion on the clinical impact of this work, any interpretability analysis on the interaction between expression and image tokens, and the utility of having a fused embedding and any advantage that would have compared to the late fusion strategies.

It would also be interesting to see if the same holds for other types of imaging + spatial transcriptomics datasets beyond H&E and Xenium core panel.

The benchmarking procedure would be helpful as a community resource if it will be made public and well-maintained.

---

### Official Review · Reviewer_nqY8 · 2026-02-25

**Rating:** 3
**Confidence:** 5

**Review:**

The authors propose an early-fusion multimodal transformer designed to integrate histopathology (H&E whole-slide images) with spatial transcriptomics (ST), specifically subcellular Xenium transcript readouts. Unlike prior CLIP-style models that rely on contrastive alignment in aggregated latent spaces, this model injects transcript-derived tokens directly into the Vision Transformer (ViT) token sequence prior to shared processing, bypassing the need for cell segmentation. The model is evaluated on a gene prediction task: predicting an "add-on" panel of 100 held-out genes given histology and a core panel of 380 genes.


### Strengths

- Insightful Ablations on Fusion Mechanisms: The most scientifically valuable contribution of the paper is the demonstration that the granularity of the transcript representation (token-level vs. tile-level) is the primary driver of performance, rather than the timing of the fusion (early vs. late). The finding that a tokenized late-fusion model performs on par with early-fusion provides excellent nuance to the multimodal architecture discourse.

### Weaknesses

- Exclusive Reliance on a Proprietary Dataset : The most glaring empirical limitation of this paper is its complete dependence on an internal, private cancer dataset of 159 patients for all experiments and evaluations. The authors do not introduce this as a new public benchmark, nor do they provide access to it, which severely compromises the reproducibility of their findings.

- Avoidance of Established Community Benchmarks : The authors are clearly aware of standardized public benchmarks, explicitly referencing the HEST-1k dataset and noting that recent large-scale evaluations of competing CLIP-based approaches have been conducted on it. However, they relegate testing their own model on HEST-1k to "future work".

- Unproven Generalizability and Robustness : Spatial transcriptomics technologies are notoriously susceptible to batch effects and variance across different tissue types and experimental setups. By training and evaluating exclusively on a single internal cohort using a highly specific gene panel (a 380-core and 100-target add-on panel), the authors fail to demonstrate that their multimodal transformer can generalize. Without a multi-center public benchmark, it remains unclear whether the observed performance gains are due to the architecture itself or simply the model overfitting to the specific artifactual nuances of their internal data pipeline.

---

### Meta-Review · Area_Chair_vYTH · 2026-02-28

**Recommendation:** Accept (Poster)
**Confidence:** 4

**Metareview:**

This paper has the largest split in opinion of all the papers in my batch. Reviewer nqY8's concerns are valid and very important to address in more complete presentations of this work, but given that this is the Tiny Paper track, the paper is worth discussing at the workshop, so I'm recommending acceptance.

---

### Decision · Program_Chairs · 2026-03-02

**Decision:**

Accept (Poster)

**Comment:**

Please see the meta-review.